# Recovery of Natural Antioxidants from Onion Solid Waste via Pressurized Liquid Extraction: Encapsulation and Application into a Food System

**DOI:** 10.3390/foods14203583

**Published:** 2025-10-21

**Authors:** Eleni Bozinou, Nafsika-Thalia Georgiadou, Maria-Stella Chalastara, Ioannis Makrygiannis, Martha Mantiniotou, Vassilis Athanasiadis, Arhontoula Chatzilazarou, Stavros I. Lalas

**Affiliations:** 1Department of Wine, Vine & Beverage Sciences, University of West Attica, Ag. Spyridonos Str., 12243 Athens, Greece; ebozinou@uniwa.gr; 2Department of Food Science and Nutrition, University of Thessaly, Terma N. Temponera Street, 43100 Karditsa, Greece; nageorgiadou@uth.gr (N.-T.G.); mchalastara@uth.gr (M.-S.C.); ioanmakr1@uth.gr (I.M.); mmantiniotou@uth.gr (M.M.); vaathanasiadis@uth.gr (V.A.)

**Keywords:** polyphenols, natural preservatives, lipid oxidation, mayonnaise, response surface methodology, waste valorization

## Abstract

Onion solid wastes (OSW) are a promising source of natural antioxidants with potential applications in food preservation. This study optimized pressurized liquid extraction parameters—ethanol concentration, liquid-to-solid ratio, temperature, and time—using response surface methodology to maximize the recovery of total polyphenols, total anthocyanins, and antioxidant activity. The optimal extracts yielded 37.02 mg GAE/g dw for total polyphenols and 592.73 µg CyE/g dw for total anthocyanins. Antioxidant activity was evaluated using the FRAP assay and the DPPH method. HPLC-DAD analysis identified quercetin, spiraeoside, and protocatechuic acid as major polyphenolic compounds. The optimal extract was encapsulated via spray drying using gum arabic as the wall material. Parameters such as encapsulation efficiency, loading capacity, and process yield were evaluated. The encapsulated extract was incorporated into mayonnaise, and its effect on oxidative stability was monitored over 14 days. Results demonstrated significant antioxidant protection, comparable to synthetic preservatives such as potassium sorbate and butylated hydroxytoluene. This study highlights the potential of OSW-derived antioxidants as sustainable, natural additives for food systems.

## 1. Introduction

Onions (*Allium cepa* L.) are one of the most important and widely cultivated vegetables globally, ranking second after tomatoes in cultivation volume. Global onion production has increased by more than 25% in recent years [1]. Current worldwide onion production is estimated at 93.23 million tons [2]. This high production generates a substantial amount of onion solid waste (OSW), which includes the semi-dried edible outer layers, the dry layers, and the apical and basal trimmings, as well as undersized, deformed, infected, or broken bulbs [3]. This waste stream creates both biological and environmental challenges. OSW is not appropriate as animal feed, cannot be used as fertilizer [2], and should not be released into landfills due to its strong sulfurous aroma and the fact that it promotes microbial growth, such as of *Sclerotium cepivorum*. Additionally, due to its high moisture content, incineration is economically unfavorable [3,4,5].

Flavonoids constitute the largest phenolic group in onions [6,7,8]. Red onions contain the highest flavonoid levels, followed by yellow onions [3]. According to Chadorshabi et al. [3], flavonoids in red onion skins range from 1.276 to 169 mg/g, and their amount is higher than that of white onion skin, with an average value of 0.08 mg/g. Flavonoids exhibit different biological activities such as antioxidant, antibacterial, antifungal, cardioprotective, anti-inflammatory, antiviral, neuroprotective, anti-obesity, anticancer, and antidiabetic effects [3].

The first step in obtaining bioactive compounds from OSW is extraction, and both conventional and modern techniques have been explored. Conventional methods such as Soxhlet and maceration are effective but require long times and large solvent volumes [9]. More advanced approaches, including microwave- and ultrasound-assisted extraction, have been optimized for quercetin recovery and improved efficiency [10,11]. Subcritical water extraction has also been applied, offering high yields under eco-friendly conditions [12]. In parallel, green solvents such as deep eutectic mixtures have shown promise for selective recovery of flavonoids [13]. Despite these advances, there are very few studies on pressurized liquid extraction (PLE), and none that combine PLE with encapsulation and direct application of OSW extracts in real food systems.

Green extraction techniques, whether applied independently or in combination with conventional methods, are increasingly recognized for their efficiently in recovering a wide range of bioactive substances [14]. Among them, PLE applies elevated temperature and pressure to enhance solvent penetration, solubility, and mass transfer, thereby accelerating extraction and improving yields [15]. When operated under near-subcritical conditions, PLE further increases efficiency while maintaining solvent safety and reproducibility [16].

PLE outperforms conventional (e.g., Soxhlet, maceration) and many modern extraction techniques by combining high pressure and temperature to achieve faster extraction, lower solvent use, and higher yields with excellent reproducibility. Studies show PLE can recover up to 30–40% of plant material in minutes versus hours and with sixfold less solvent than Soxhlet, while outperforming supercritical CO_2_ and ultrasound/microwave-assisted methods in efficiency and consistency [15,17,18,19,20]. Although equipment cost and risk of thermolabile degradation are drawbacks, PLE’s balance of speed, efficiency, and scalability makes it superior for routine and industrial applications [21]. As a result, PLE is regarded as a high-performance and eco-friendly technique for extracting bioactive compounds from natural sources, such as plants, offering broad potential applications in the food industry [22].

The rising demand for plant-based products rich in bioactive compounds has increased the need for methods that enhance food stability and shelf life. To achieve this, plant extracts are often converted into powders using spray drying—a process that rapidly removes moisture by exposing the material to hot air, producing a stable powder with preserved physicochemical properties [23]. This method also encapsulates sensitive compounds, protecting them from light, moisture, and oxygen [24,25]. Spray drying improves product stability, quality, and shelf life while reducing volume and weight, making storage and transport easier [23]. The final product quality depends on the properties of the feed solution, spray dryer settings, and the type and ratio of carriers used [26]. Common carriers include polysaccharides, proteins, and lipids, with β-cyclodextrin (β-CD), gum arabic (GA), and maltodextrin (MD) widely used for their solubility, low viscosity, affordability, and stabilizing properties [23,27].

The aim of this study is to develop and evaluate the potential use of OSW encapsulated extracts as natural antioxidants in food. The optimal conditions for PLE of OSW were identified with the use of response surface methodology (RSM). Key extraction parameters—solvent composition (ethanol–water), temperature, liquid-to-solid ratio, and extraction time—were optimized. The optimal extract was analyzed for total polyphenol content (TPC) and total anthocyanin content (TAC), as well as antioxidant activity using DPPH^•^ and FRAP assays. Moreover, the optimized extract was encapsulated by spray drying in GA and analyzed for encapsulation efficiency (%), loading capacity (%), and process yield (%). High-performance liquid chromatography (HPLC) with diode-array detection (DAD) was applied to the resulting powder for the identification of individual polyphenolic compounds. Finally, the encapsulated extract was incorporated into a food system (mayonnaise), and oxidative stability was monitored over 14 days. According to the authors’ knowledge, no prior study has integrated PLE optimization, encapsulation, and direct application of OSW extracts in a real food system (mayonnaise).

## 2. Materials and Methods

### 2.1. Chemicals and Reagents

Polyphenolic standards of HPLC grade (≥99.0% *w*/*w*) were obtained from MetaSci (Toronto, ON, Canada). From Panreac (Barcelona, Spain), gallic acid (≥99.0% *w*/*w*), Folin–Ciocalteu reagent, ammonium iron(II) sulfate hexahydrate (≥99.0% *w*/*w*), ethanol (≥99.8% *v*/*v*), and thiobarbituric acid were sourced. Trolox (≥96.5% *w*/*w*) was purchased from Glentham Life Sciences (Corsham, UK). Sigma-Aldrich (Darmstadt, Germany) supplied TPTZ (≥98% *w*/*w*), DPPH (≥90.0% *w*/*w*), methanol (≥99.8% *v*/*v*), hydrochloric acid (37% *w*/*w*), and trichloroacetic acid (≥99.0% *w*/*w*). Iron(III) chloride hexahydrate (≥99.0% *w*/*w*) was obtained from Merck (Darmstadt, Germany). Penta (Prague, Czech Republic) provided ammonium thiocyanate (≥99.0% *w*/*w*), chloroform, formic acid (99.8%), and anhydrous sodium carbonate. Acetonitrile (99.9%) was purchased from Labkem (Barcelona, Spain), while dichloromethane and ethyl acetate were obtained from Carlo Erba (Vaulx-de-Reuil, France). Hydrogen peroxide (35%) was sourced from Chemco (Malsch, Germany). Deionized water used in all experiments was produced using a deionizing column.

### 2.2. Onion Solid Waste (OSW) Material

The red onions used for this study were purchased from a local market in Karditsa, Greece. The variety was “MIRSINI”, with a diameter greater than 75 mm, and the cultivation region was Viotia, Greece. The OSW (skins, outer fleshy scales, and apical and basal trimmings) were removed and immediately subjected to freeze-drying. The freeze-drying process was carried out using a BK-FD10P lyophilizer (Biobase, Jinan, China). Following drying, the material was ground using an electric milling device to reduce particle size, thus increase surface area, mass transfer and solvent penetration for more efficient extraction. The ground material was then sieved using an Analysette 3 PRO sieve shaker (Fritsch GmbH, Idar-Oberstein, Germany), resulting in a median particle size of 144 μm. The resulting powder was stored at −40 °C until used in the subsequent experiments.

### 2.3. Experimental Design

To optimize the extraction of total polyphenol content (TPC), total anthocyanin content (TAC), and antioxidant activity (measured via FRAP and DPPH^•^ assays) from OSW powder, a Response Surface Methodology (RSM) approach was applied using a Custom Quadratic design. This design, chosen for its efficiency and rapid data acquisition, allowed for thorough exploration of four key factors, each evaluated at three levels. A Pressurized Liquid Extraction (PLE) system (Fluid Management Systems, Inc., Watertown, MA, USA) was used to perform all extractions under a constant pressure of 1700 psi, as determined from preliminary testing. The independent variables included the solvent composition, specifically the aqueous ethanol concentration (*C*, % *v*/*v*), reflecting solvent polarity (*X*_1_); the liquid-to-solid ratio (*R*, mL/g) (*X*_2_), which is equivalent to the solvent-to-feed ratio (S/F) commonly used in extraction engineering; the extraction temperature (*T*, °C) (*X*_3_); and the extraction time (*t*, min) (*X*_4_). Each factor was tested at three coded levels: low (−1), medium (0), and high (+1), as shown in Table 1. Solvent levels were selected to span a polarity gradient, while the remaining factor levels were established based on prior experimental results.

To ensure method reproducibility, 18 experiments were conducted—each one repeated three times and incorporating three central points. Mean values of all responses were computed for analysis. To enhance model accuracy, a stepwise regression approach was applied, removing non-essential terms to reduce variance. This yielded a second-order polynomial equation representing the effects and interactions of the independent variables:(1)$$Y_{k}\;=\;\beta_{0}+\sum_{i=1}^{2}\beta_{i}X_{i}+\sum_{i=1}^{2}\beta_{ii}X_{i}^{2}+\sum_{i=1}^{2}\sum_{j=i+1}^{3}\beta_{ij}X_{i}X_{j}$$
where *Y_k_* is the predicted response, while *X_i_* and *X_j_* denote the independent variables. The coefficients *β*_0_, *β_i_*, *β_ii_*, and *β_ij_* correspond to the intercept, linear terms, quadratic terms, and interactions, respectively.

### 2.4. Determination of Antioxidant Components by Spectrophotometric Methods

#### 2.4.1. Total Polyphenolic Content (TPC)

The total phenolic content (TPC) was assessed using the Folin–Ciocalteu method [28], with results reported as milligrams of gallic acid equivalents (GAE) per gram of dry weight (dw). A calibration curve ranging from 10 to 100 mg/L of gallic acid (R^2^ = 0.9996) in water was employed for quantification. Briefly, 100 μL of the appropriately diluted extract was mixed with 100 μL of Folin–Ciocalteu reagent and allowed to react for 2 min. Next, 800 μL of a 5% *w/v* sodium carbonate solution was added. The mixture was incubated for 20 min at 40 °C, shielded from light, and absorbance was then measured at 740 nm using a Shimadzu UV-1900i UV/Vis spectrophotometer (Kyoto, Japan). The 40 °C incubation was carried out using an Elmasonic P70H ultrasonic bath from Elma Schmidbauer GmbH (Singen, Germany). Each sample was analyzed in triplicate, and the average result was used for calculations.

#### 2.4.2. Total Anthocyanin Content (TAC)

The total anthocyanin content (TAC) was determined using a previously described method [29]. A 70 μL aliquot of the extract was combined with 930 μL of a hydrochloric acid solution (0.25 M in ethanol) in a 1.5 mL Eppendorf tube and vortexed. After 10 min, the absorbance was measured at 520 nm, with the ethanolic HCl solution serving as the blank. The concentration of total anthocyanins (*C*_TA_) was then calculated as cyanidin-3-*O*-glucoside equivalents (CyE) using Equation (2):(2)$$C_{\mathrm{TA}}\;(\mathrm{mg}\;\mathrm{CyE}/L)\;=\;\frac{A\;\times \;\mathrm{MW}\;\times \;F_{D}\;}{\varepsilon }\;\times \;10^{3}$$
where *A* is the absorbance at 520 nm, MW is the cyanidin-3-*O*-glucoside molecular weight (449.2), *F*_D_ is the dilution factor, and ε = 26,900.

Therefore, the TAC was determined as follows in Equation (3):(3)$$\mathrm{TAC}\;(\mathrm{mg}\;\mathrm{CyE}/g\;\mathrm{dw})\;=\;\frac{C_{\mathrm{TA}}\;\times \;V}{w}$$
where *V* denotes the volume of the extraction solvent (in L), and *w* represents the dry weight of the sample (in g).

#### 2.4.3. Ferric-Reducing Antioxidant Power (FRAP, P_R_) Activity

A previously established method, based on the common electron-transfer technique, was used to assess the antioxidant capacity of the extracts [28]. This approach involved measuring the reduction in the iron oxidation state from +3 to +2. Briefly, 50 μL of the appropriately diluted sample was combined with 50 μL of FeCl_3_ solution (4 mM in 0.05 M HCl) and incubated at 37 °C for 30 min. After 5 min, 900 μL of TPTZ solution (1 mM in 0.05 M HCl) was added, and the absorbance was recorded at 620 nm. Quantification was performed using a calibration curve prepared with ascorbic acid (50–500 μM in 0.05 M HCl, R^2^ = 0.9997). The results were expressed as μmol of ascorbic acid equivalents (AAE) per gram of dry weight (dw). All measurements were carried out in triplicate, and mean values were reported.

#### 2.4.4. DPPH^•^ Radical Scavenging Activity

The DPPH^•^ scavenging assay, as described in a previous study [28], was applied. To begin, 25 μL of the appropriately diluted sample extract was mixed with 975 μL of DPPH^•^ solution (100 μmol/L in methanol), and absorbance at 515 nm was recorded immediately and after 30 min. Quantification was based on a calibration curve constructed with ascorbic acid (100–1000 μmol/L in methanol, R^2^ = 0.9926). Results were expressed as μmol of ascorbic acid equivalents (AAE) per gram of dry weight (dw). All measurements were carried out in triplicate, and mean values were reported.

### 2.5. Spray Drying Encapsulation

Prior to spray drying, the optimized onion solid waste (OSW) extract was mixed with gum arabic (GA) as the encapsulating agent. Since the optimal extract was obtained using ethanol, solvent removal was first performed with a Heidolph Laborota 4000/G3 rotary evaporator, equipped with Rotavac Valve Control (Heidolph Instruments GmbH & Co. KG, Schwabach, Germany). The resulting concentrate was resuspended in a 60:40 (*v*/*v*) ethanol–water solution, followed by a second evaporation step to ensure complete removal of ethanol. The remaining aqueous extract was subsequently mixed with GA at a sample-to-carrier ratio of 1:6 (*w*/*w*), determined based on the total polyphenol content (TPC) of the resuspended extract.

The spray drying process was carried out using a BUCHI mini-B-290 laboratory spray dryer (BUCHI, Flawil, Switzerland) equipped with a standard 1-mm nozzle. Operating parameters were adapted with slight modifications from the method described by [30]. The inlet and outlet air temperatures were set to 170 °C and 95 °C, respectively. The solution was fed at a rate of 3.5 mL/min, with a spray air flow rate of 742 L/h and a pressure drop of 1.35 bar. The aspirator was operated at a gas flow rate of 35 m^3^/h. After spray drying, the resulting powders were stored in sealed plastic containers and kept refrigerated until further analysis.

### 2.6. Microcapsule Characterization Metrics

#### 2.6.1. Process Yield

The yield obtained from the spray-drying process was determined in accordance with Equation (4) [31]:(4)$$\mathrm{Process}\;\mathrm{yield}\;(\%)\;=\;\frac{m_{p}}{m_{d}\;+\;m_{c}}\;\times \;100$$
where *m*_p_ is the mass of the powder obtained (g), *m*_d_ is the dry matter of the extract in the volume used for drying (g), and *m*_c_ is the mass of carrier (g) incorporated into the extract prior to spray-drying.

#### 2.6.2. Encapsulation and Loading Capacity

Encapsulation capacity was evaluated as the proportion of total to surface polyphenols in the microcapsules, following the procedure of Robert et al. [32], while loading capacity was expressed as the amount of polyphenols retained in the microcapsules relative to their dry weight after spray drying [33].

To determine total polyphenols (TP), 0.2 g of powder was mixed with 2 mL of a methanol–acetic acid–water solution (50:8:42, *v*/*v*/*v*), vortexed for 1 min, and subjected to ultrasonic extraction at room temperature for 20 min. The extract was then centrifuged at 3000 rpm for 10 min, and the supernatant was analyzed using the Folin–Ciocalteu method [23].

Surface polyphenols (SP) were obtained by mixing 0.2 g of powder with 2 mL of ethanol–methanol (50:50, *v*/*v*), followed by vortexing for 1 min and centrifugation at 3000 rpm for 10 min. The supernatant was filtered and analyzed in the same way as TP.

Encapsulation capacity (EC) was calculated according to Equation (5):(5)$$\mathrm{EC}\;\left(\%\right)\;=\;\left(\frac{\mathrm{TP}\;-\;\mathrm{SP}}{\mathrm{TP}}\right)\;\times \;100$$
where TP is the concentration of total polyphenols (mg GAE/g) and SP is the concentration of surface polyphenols (mg GAE/g).

Loading capacity (LC) was determined using Equation (6):(6)$$\mathrm{LC}\;(\%)\;=\;\frac{\mathrm{TP}}{\mathrm{MC}}\;\times \;100$$
where TP is the total polyphenols in the microcapsules (g), and MC is the microcapsule weight (g) after spray drying.

#### 2.6.3. HPLC-DAD Quantification of Key Polyphenols

The analysis was conducted using a Shimadzu CBM-20A liquid chromatograph coupled with a Shimadzu SPD-M20A diode array detector (Shimadzu Europa GmbH, Duisburg, Germany), following the procedure described in a previous study [28]. Separation was performed on a Phenomenex Luna C18 (2) column (100 Å, 5 μm, 4.6 × 250 mm; Phenomenex Inc., Torrance, CA, USA) maintained at 40 °C. The mobile phase consisted of (A) 0.5% aqueous formic acid and (B) 0.5% formic acid in acetonitrile/water (6:4, *v*/*v*). The gradient elution was programmed as follows: 0% B at the start, increased to 40% B, then to 50% B within 10 min, further raised to 70% B over the next 10 min, and maintained at 70% B for 10 min. The mobile phase flow rate was 1 mL/min. Compounds were identified by comparing their retention times and UV–Vis spectra with those of authentic standards. Quantification was based on calibration curves prepared with concentrations ranging from 0 to 50 μg/mL.

### 2.7. Integration into Mayonnaise Matrix

#### 2.7.1. Mayonnaise and Mayonnaise Samples’ Preparation

Mayonnaise was prepared according to the formulation presented in Table 2, using sunflower oil, olive oil, egg yolk, mustard, vinegar, lemon juice, salt, and pepper powder. Following preparation, different concentrations of the encapsulated OSW extract were incorporated into 100 g portions of mayonnaise, which were placed in sterile plastic containers. A 100 g sample size was selected to ensure homogeneous mixing and reproducible incorporation of encapsulated onion solid waste (EOSW) without compromising analytical accuracy. Each treatment was prepared in quintuplicate, providing five independent replicates per condition. This ensured reproducibility and sufficient material for repeated analyses across all time points. In addition, a post hoc power analysis of the main oxidative stability parameters (peroxide value and TBARS) confirmed that the selected sample size achieved statistical power > 0.8 at α = 0.05, demonstrating adequacy for detecting treatment effects. For comparison, two synthetic antioxidants—butylated hydroxytoluene (BHT) and potassium sorbate—were added at different concentrations (Table 3). The concentrations used are based on preliminary experiments and the European (EU) Union Regulations and Greek Food and Beverage Code for mayonnaise. A control sample consisting of mayonnaise without antioxidant addition was also prepared. Each sample was prepared in quintuplicate, so as one sample could be used for each testing day.

All samples were stored at ambient temperature (25 ± 2 °C) in the dark for 14 days. On days 1, 3, 7, and 14, aliquots were withdrawn and analyzed to determine the oxidative stability of the formulations.

#### 2.7.2. Determination of Oxidative Stability Parameters

At each sampling point, the pH and color of the mayonnaise samples were measured. Antioxidant activity and oxidative stability were further evaluated in both the polar and non-polar fractions. In the polar fraction, analyses included the determination of total polyphenol content (TPC) and DPPH^•^ radical scavenging activity. In the non-polar fraction, DPPH^•^ activity, peroxide value (PV), and thiobarbituric acid reactive substances (TBARS) formation were assessed.

The pH value of the samples was measured with an XS pH 60 VioLab Bench pH meter coupled with a 201 T DHS electrode (Capri, Italy). The color of the extracts was assessed using a method previously established by Cesa et al. [34]. The CIELAB parameters (*L**, *a**, and *b**) were measured for the hydroalcoholic extracts with a colorimeter (Lovibond CAM-System 500, The Tintometer Ltd., Amesbury, UK). The color description is based on three parameters: The *L** value represents lightness, ranging from 0 (absolute black) to 100 (absolute white). The *a** value indicates the degree of greenness (positive values) or redness (negative values). Similarly, the *b** value reflects the extent of blueness (positive values) or yellowness (negative values). Color intensity is represented by *C_ab_* or *C** (Chroma, saturation), which is calculated using the equation C* = (*a*^2^ + *b*^2^)^(1/2)^.

#### 2.7.3. Polar Compound Determinations

One gram of mayonnaise was mixed with 2 mL of *n*-hexane to dissolve the lipid fraction, followed by extraction with 2 mL of a methanol–water solution (60:40 *v*/*v*). The mixture was vortexed thoroughly and centrifuged at 4500 rpm for 5 min. The resulting polar (hydrophilic) extract was then collected and used for total polyphenol and DPPH^•^ antioxidant analyses. The total polyphenol and DPPH^•^ protocols are described in Section 2.4.1 and Section 2.4.4, respectively.

#### 2.7.4. Non-Polar Compound Determinations

For the determination of non-polar compounds, 1.5 g of mayonnaise was placed in a 2 mL Eppendorf tube and subjected to centrifugation at 10,000 rpm. This process facilitated the separation of the oil phase, which was subsequently collected for DPPH^•^, peroxide value (PV), and thiobarbituric acid reactive substances (TBARS) assays.

The radical scavenging activity of oils was evaluated using the DPPH^•^ assay in the lipophilic phase, following the procedure of [33]. A 950 μL aliquot of DPPH^•^ solution (100 μM in ethyl acetate) was combined with 50 μL of oil diluted tenfold in ethyl acetate. Reduction of the DPPH^•^ radical to its non-radical form was monitored spectrophotometrically, and antioxidant capacity was quantified against a Trolox calibration curve (50–500 μM, R^2^ = 0.999). Results were expressed as Trolox equivalent antioxidant capacity (TEAC, μM/kg oil).

Lipid peroxidation was assessed via peroxide value determination according to IDF method 74A:1991, with minor modifications [33]. This method quantifies hydroperoxides through their reaction with Fe^2+^ and thiocyanate, yielding a red Fe^3+^–thiocyanate complex. For analysis, 20 μL of oil was dissolved in 1960 μL dichloromethane/ethanol (3:2, *v*/*v*), followed by the addition of 10 μL each of ammonium thiocyanate and ammonium iron (II) sulfate. After 5 min, absorbance was recorded at 500 nm. A hydrogen peroxide calibration curve (50–500 μM, R^2^ = 0.995) was used for quantification, and values were expressed as mmol H_2_O_2_/kg oil.

Secondary oxidation products were determined by the TBARS assay [35]. Oil samples (0.1 g) were incubated with 5 mL of TBA reagent (15 g trichloroacetic acid, 1.76 mL HCl, and 0.375 g TBA in 100 mL water) at 95 °C for 20 min. After cooling, 200 μL chloroform was added, and the mixture was centrifuged (4500 rpm, 10 min). The absorbance of the supernatant was measured at 532 nm, using water as the blank. Values were expressed as mmol malondialdehyde equivalents (MDAE)/kg oil, calculated from a calibration curve (15–300 μM, y = 0.0032x − 0.0004, R^2^ = 0.9999).

### 2.8. Statistical Analysis

All statistical analyses were performed using JMP^®^ Pro 16 (SAS Institute Inc., Cary, NC, USA). Data were first tested for normality using the Kolmogorov–Smirnov test. Differences among treatments were evaluated by analysis of variance (ANOVA), and means were compared using Tukey’s HSD test at a significance level of *p* < 0.05. Each extraction experiment was conducted in duplicate, and all measurements were obtained in triplicate. Results are expressed as mean ± standard deviation. Multivariate techniques, including partial least squares (PLS), principal component analysis (PCA), multiple correspondence analysis (MCA), and Pareto plot analysis, were further applied to explore variable relationships and identify the most influential extraction parameters.

## 3. Results and Discussion

### 3.1. Optimization of PLE Parameters

Custom Quadratic design via RSM was applied to evaluate the influence of individual extraction factors and to determine the optimal extraction conditions. The effect of each variable was assessed based on the outcomes of the respective assays, and the results are summarized in Table 4. Experimental designs 1 and 17 were the most effective for the extraction of polyphenols and anthocyanins, with correspondingly high antioxidant activities, while the least effective combination was design point 4. Taking the results into account, the most appropriate solvent was pure ethanol, while the lowest responses were obtained with water. Regarding temperature, the highest tested value (160 °C) proved to be the most favorable for the liquid-to-solid ratios of 70 and 10 mL/g. The same pattern was observed for extraction time. The liquid-to-solid ratio did not show a consistent pattern, as it differed between the two most favorable design points, indicating that the combined effect of all tested parameters significantly influenced the extraction of bioactives. Although both liquid-to-solid ratio (equivalent to solvent-to-feed ratio, S/F) and extraction time were included in the RSM model to capture potential interactions, the analysis confirmed that the L/S (S/F) ratio was the dominant factor influencing extraction efficiency. For practical and industrial applications, the L/S (S/F) ratio is therefore the decisive parameter, while extraction time plays a secondary role.

The choice of solvent is a critical factor that significantly affects extraction efficiency [36]. Bioactive compounds such as polyphenols, which have moderate polarity, are not effectively extracted using water alone due to its high polarity [29]. However, using ethanol mixed with water enhances the extraction process, as the solvent combination better matches the polarity of the target compounds [37]. Temperature also plays a vital role in optimizing extraction, as it can greatly influence yield, particularly for polyphenols [38]. High temperatures can promote the deglycosylation of quercetin, leading to extracts with higher antioxidant activity, since quercetin itself exhibits greater antioxidant action than its glycosylated forms [39,40].

The high degree of correlation among sample values is presented in Table 5, alongside the statistical outputs of the stepwise ANOVA regression analyses conducted for the assays. The final regression model excludes variables that did not exhibit statistical significance (*p* > 0.05). Notably, the model demonstrates an excellent fit to the experimental data, as evidenced by the high coefficient of determination (R^2^ > 0.96 in most cases), indicating strong explanatory power.

### 3.2. Model Analysis

The mathematical models for each assay, created using Custom Quadratic design via RSM, are shown in Equations (7–10). They are second-order polynomial equations containing only statistically significant terms. Table 6 presents the maximum predicted responses and optimum extraction conditions for the dependent variables. High desirability values (>0.91) were obtained for all assays. TPC and TAC extraction required different conditions to achieve maximum values, whereas both FRAP and DPPH assays shared identical optimal conditions, which differed from those for TPC and TAC. High-polarity solvents (76–100% ethanol) were required for efficient extraction of TPC and TAC, as well as for maximum antioxidant activity. The optimum temperature was the highest tested (160 °C), indicating that such high heat enhances mass transfer and extractability, thereby maximizing yields. For the liquid-to-solid ratio, the lowest tested value (10 mL/g) was optimal for antioxidant activity, whereas higher ratios were needed to maximize TPC and TAC.*TPC* = 10.66 + 0.30*X*_1_ + 0.14*X*_2_ − 0.44*X*_3_ + 0.19*X*_4_ − 0.001*X*_1_^2^ − 0.002*X*_2_^2^ + 0.002*X*_3_^2^ − 0.017*X*_4_^2^ − 0.002*X*_1_*X*_2_ + 0.001*X*_1_*X*_3_ − 0.013*X*_1_*X*_4_ + 0.007*X*_2_*X*_4_ + 0.011*X*_3_*X*_4_(7)*TAC* = 223.43 + 0.39*X*_1_ − 4.95*X*_2_ − 0.12*X*_3_ + 0.090*X*_2_^2^ + 0.013*X*_1_*X*_3_(8)*FRAP* = 113.62 + 0.06*X*_1_ − 0.48*X*_2_ − 2.49*X*_3_ − 2.19*X*_4_ + 0.007*X*_1_^2^ + 0.024*X*_2_^2^ + 0.012*X*_3_^2^ − 0.017*X*_1_*X*_2_ + 0.010*X*_1_*X*_3_ − 0.006*X*_2_*X*_3_ + 0.048*X*_3_*X*_4_(9)*DPPH* = 32.16 + 0.22*X*_1_ + 0.02*X*_2_ − 0.79*X*_3_ − 0.55*X*_4_ + 0.008*X*_2_^2^ + 0.004*X*_3_^2^ − 0.006*X*_1_*X*_2_ + 0.003*X*_1_*X*_3_ − 0.002*X*_2_*X*_3_ + 0.014*X*_3_*X*_4_(10)

Although the Custom Quadratic design via RSM ranged between 0–100% ethanol, 10–70 mL/g liquid-to-solid ratio, 40–160 °C temperature, and 5–25 min time, the desirability profiler predicted an optimum at 100% EtOH, 70 mL/g ratio, 160 °C and 25 min—technically the maximum examined values. For verification purposes, three independent extractions ran under those exact conditions. The experimentally measured values (Table 7)—TPC = 37.02 ± 1.18 mg GAE/g dw (predicted 34.75); TAC = 592.73 ± 31.41 μg CyE/g dw (predicted 540.88), FRAP = 291.15 ± 17.76 μmol AAE/g dw (predicted 317.22); DPPH = 98 ± 6.57 μmol AAE/g dw (predicted 106.63)—all fall very close to the model’s predictions. This close agreement confirms that the second-order polynomial equations demonstrate an excellent fit to the experimental data. The Pareto plot analysis (Appendix A) as well as PCA (Appendix A) and MCA (Appendix A) are provided in the Appendix A. Appendix A shows that the correlations among the variables were generally positive and relatively strong, with values ranging from 0.5082 (TAC vs. TPC) to 0.988 (FRAP vs. DPPH).

### 3.3. Partial Least Squares (PLS) Analysis

The effects of extraction parameters (*X*_1_–*X*_4_) were evaluated using a partial least squares (PLS) model, with the correlation loading plot presented in Figure 1A. This analysis highlighted the influence of each factor on OSW extracts. Solvent composition (*X*_1_) exerted the strongest effect, with 100% ethanol yielding the highest responses across assays. The liquid-to-solid ratio (*X*_2_) of 70 was identified as optimal for efficient extraction. Temperature (*X*_3_) also showed a positive contribution, with 160 °C being the most favorable condition. Finally, extraction time (*X*_4_) demonstrated a positive association with responses, indicating that longer durations (25 min) were generally more effective.

Figure 1B illustrates the relative importance of the extraction parameters on bioactive compound levels and antioxidant activity. Solvent concentration was identified as the most influential factor, with an importance value of ~1.58, well above the threshold of 0.8. Extraction temperature also exerted a strong effect, with an importance value of 1.48. These findings underscore the pivotal role of solvent concentration and temperature in maximizing the bioactive composition and antioxidant capacity of the extracts.

The experimental findings align exceptionally well with the PLS model predictions, as demonstrated by the impressive correlation coefficient of 0.992 and a strong coefficient of determination (R^2^) of 0.984. Additionally, the *p*-value below 0.0001 suggests that the differences between observed and predicted values are not statistically significant.

Table 7 shows the optimal extract yields, which reached 37.02 ± 1.18 mg GAE/g dw for polyphenols and 592.73 ± 31.41 µg CyE/g dw for anthocyanins. Antioxidant activity was assessed using the FRAP assay showing a value of 291.15 ± 17.76 µmol AAE/g dw, followed by the DPPH method with a value of 98 ± 6.57 µmol AAE/g dw.

### 3.4. Encapsulation and Characterization Metrics

The optimal extract was encapsulated via spray drying using GA as the wall material, achieving 94.64% encapsulation efficiency, 2.31% loading capacity, and 84.03% process yield. The encapsulated extract (after proper treatment) was subjected to HPLC analysis for the identification of the individual polyphenols present to the powder. Table 8 presents the Individual polyphenolic compounds of OSW, followed by encapsulation in GA. It also includes the calibration curve equations for each compound as determined by HPLC-DAD analysis. Figure 2 presents the HPLC-DAD analysis, where quercetin, spiraeoside (quercetin 4′-*O*-glucoside), and protocatechuic acid were identified as major polyphenolic compounds. These findings are in line with literature. Specifically, Bozinou et al. [9,39,41,42] studied the extraction of OSW using cyclodextrins as co-solvents with water, as well as deep eutectic solvents and aqueous ethanol solvents and found, among others, quercetin, spiraeoside and protocatechuic acid as major polyphenolic constituents of OSW. In another study where the extraction of phenolics from onion and garlic residues were investigated, quercetin and protocatechuic acids were also identified [43]. Rguez et al. [44] studied the extraction of quercetin from *Allium cepa* using vinegar. The phytochemical profile of the extracts consisted of quercetin and quercetin derivatives, in particular quercetin 4′-*O*-β-glycoside, quercetin 3-*O*-β-glucoside and quercetin 3,4′-*O*-β- diglucoside, confirming once again the polyphenolic profile found in the current study.

### 3.5. Mayonnaise Matrix

Different concentrations of encapsulated OSW extract (EOSW) at 0.05, 0.1, and 0.5% *w*/*w* were incorporated into mayonnaise, and their effect on oxidative stability was monitored over a period of 14 days. Multiple parameters were evaluated across four sampling days (Days 1, 3, 7, and 14), including pH, color, and TPC, TAC, DPPH and FRAP for the polar fraction, as well as PV, TBARS and DPPH for the non-polar (lipophilic) fraction of the mayonnaise.

#### 3.5.1. Oxidative Stability Results

As can be seen in Table 9, the pH of the samples remained remarkably stable over the 14-day period. No sample showed significant day-to-day variation (small letters “a” across Days 1, 3, 7 and 14), indicating pH stability within each treatment. While all samples held their pH steady over time, the EOSW at 0.50% consistently had the lowest acidity and potassium sorbate at 1000 ppm the highest, with convergence among other antioxidants by Day 3. EOSW-enriched mayonnaises-maintained pH stability on par with the control, with a slight, dose-dependent acidification (lowest pH at 0.50% EOSW) that did not disrupt storage-day equilibrium—showing the extract can subtly lower acidity without compromising overall pH stability.

An analysis of color changes over the 14-day storage period revealed a general decrease in lightness (*L**) and an increase in Chroma (*C**) across all samples. The most effective color retention was observed in the sample containing 100 ppm BHT and 0.05% EOSW, as evidenced by the smallest increase in *C** and the least reduction in *L**, respectively. As shown in Table 10, the control sample and those treated with low concentrations of potassium sorbate (250–500 ppm) exhibited the poorest color stability, with the most pronounced decline in *L** and rise in *C**. Overall, samples treated with higher antioxidant concentrations tended to exhibit increased browning, as indicated by greater shifts in both *L** and *C** values. In conclusion, encapsulated OSW extract at 0.50% and mid-range BHT doses offer the best preservation of lightness and minimal Chroma development during storage.

#### 3.5.2. Polar Compounds in Mayonnaise Samples

Both total phenolic content (TPC) and DPPH^•^ radical scavenging activity generally declined from Day 1 to Day 3, peaked around Day 7 across most treatments, and then declined by Day 14. The mid-storage spike likely reflects a transient release or formation of antioxidant-active compounds, potentially due to the hydrolysis of bound phenolics or emulsion matrix disruption, facilitating the liberation of previously sequestered phenolics [45]. Moreover, the varying behavior between Days 7 and 14—where some treatments maintained elevated activity while others dropped sharply—underscores differences in the stability and reactivity of antioxidant compounds under differing conditions. Additionally, assay interference by degradation products, such as Maillard reaction products, may artificially inflate antioxidant capacity measurements [46]. Finally, extract solubility and partitioning dynamics may contribute to delayed antioxidant peaks; lower concentrations (e.g., 0.05% EOSW) may require time to migrate into the oil phase and become detectable in assays [47].

Over a 14-day period, the total polyphenolic content (TPC) across various treatments exhibited a consistent decline from Day 1 to Day 3, suggesting an initial degradation of polyphenols, likely due to oxidation or other environmental factors. Among the treatments, EOSW 0.10% demonstrated the highest retention, with a reduction of only −53 mg GAE/kg by Day 14, indicating its superior capacity to preserve polyphenolic content. Both EOSW 0.05% and BHT 100 ppm reached a peak in TPC at Day 7, which reflects a delayed release or slower degradation of phenolic compounds. By Day 14, the control and BHT 100 ppm groups exhibited a rebound effect, stabilizing around 223–224 mg GAE/kg, while all treatments converged within a narrow range of 170–225 mg GAE/kg (Table 11). These findings suggest that while the initial decline in TPC varied between treatments, all approaches ultimately reached similar levels of polyphenolic content by the end of the study, with EOSW 0.10% demonstrating the most effective retention over the 14-day period.

Over the 14-day period, EOSW 0.10% exhibited the highest DPPH^•^ radical scavenging activity at all time points (340 → 290.8 → 401.1 → 305.9 μmol AAE/kg mayonnaise), demonstrating the smallest net decrease in antioxidant activity over the study period. All treatments showed a significant increase in antioxidant activity from Day 3 to Day 7, peaking at Day 7, followed by a decline by Day 14. Notably, EOSW 0.05% showed the largest increase in activity between Day 3 and Day 7 (+212.7 μmol AAE/kg), suggesting a strong delayed release effect. The control and potassium sorbate 250 ppm treatments exhibited the steepest declines in DPPH^•^ activity from Day 7 to Day 14 (−133 and −155 μmol AAE/kg, respectively). By Day 14, DPPH^•^ values across all treatments converged, ranging from approximately 142 μmol AAE/kg for potassium sorbate 250 ppm to 306 μmol AAE/kg for EOSW 0.10%. Among the synthetic antioxidants, BHT 200 ppm ranked second, peaking at 322.6 μmol AAE/kg on Day 7 and maintaining moderate activity by Day 14 (Table 12). These results suggest that EOSW 0.10% was the most effective treatment in terms of overall antioxidant activity, with a delayed but sustained effect observed in EOSW 0.05%, while BHT 200 ppm provided a moderate but consistent performance over time.

This dataset convincingly shows that encapsulated OSW extracts, especially at 0.10% *w/w*, outperform controls and synthetics in sustaining radical-scavenging power. The mid-storage surge is a fascinating phenomenon worth deeper investigation.

#### 3.5.3. Non-Polar Compounds in Mayonnaise Samples

All samples exhibited a time-dependent decrease in DPPH^•^ radical-scavenging capacity, indicating the consumption of antioxidants over the 14-day period. The control group demonstrated a significant decline, losing approximately 40% of its initial capacity by Day 14, decreasing from 1202 μmol TEAC/kg to 722 μmol TEAC/kg. In contrast, the EOSW at both 0.10% and 0.50% concentrations retained significantly more activity. Specifically, 0.10% EOSW preserved about 71% of the Day 1 radical-scavenging capacity by the Day 14, while 0.50% EOSW maintained approximately 80% of its initial activity, outperforming 50 ppm BHT, which retained around 65%. Notably, BHT at 100–200 ppm demonstrated the highest radical-scavenging power, remaining above 1500 μmol TEAC/kg on Day 14 (Table 13). These results indicate that EOSW, particularly at the 0.50% concentration, is an effective antioxidant with sustained activity, while BHT maintained the strongest radical-scavenging capacity overall.

As it can be seen in Table 14 and Table 15, the temporal trends in peroxide value (PV) and TBARS results together illustrate the sequential nature of lipid oxidation and the differing antioxidant efficacies of the tested treatments. In the control group, PV rose sharply from 62 to 112 μmol H_2_O_2_/kg over 14 days, followed by a corresponding increase in TBARS from 0.36 to 0.90 μmol MDAE/kg, confirming the progression from primary to secondary oxidation products. In contrast, EOSW supplementation markedly slowed both stages of oxidation. The 0.10% EOSW treatment maintained a stable PV around 67 μmol H_2_O_2_/kg throughout the 14-day period and limited TBARS accumulation to below 0.60 μmol MDAE/kg, indicating that its phenolic constituents effectively scavenged lipid radicals and stabilized peroxides, thereby delaying their breakdown. Similarly, 0.50% EOSW maintained PV below 90 μmol H_2_O_2_/kg and TBARS under 0.74 μmol MDAE/kg, demonstrating strong, dose-dependent antioxidant activity. BHT (100–200 ppm) provided the most potent protection, maintaining PV near 4–5 μmol H_2_O_2_/kg and TBARS below 0.65 μmol MDAE/kg, consistent with its well-known radical-chain-terminating mechanism. Conversely, potassium sorbate paralleled the control trend for both PV and TBARS, indicating negligible inhibition. Collectively, these correlated results confirm that EOSW and BHT effectively suppress both primary and secondary lipid oxidation, with BHT exerting the strongest overall effect.

At the highest OSW concentration, the reduced inhibitory effect—and apparent pro-oxidant behavior—may reflect concentration-dependent redox activity of phenolic compounds. At low levels, phenolics scavenge radicals, but at high concentrations they can auto-oxidize or redox-cycle, generating semiquinone radicals and hydrogen peroxide that promote lipid oxidation. Additionally, phenolics may reduce Fe(III) or Cu(II) to more reactive Fe(II)/Cu(I) forms, enhancing Fenton-type reactions. High OSW levels could also alter the emulsion’s interfacial structure and antioxidant partitioning, shifting reactive species toward the oil–water interface where oxidation is most active. Furthermore, excess phenolics can interfere with other antioxidants or emulsifiers, disrupting synergistic interactions and reducing overall protection. These combined effects likely explain the apparent pro-oxidant response at elevated OSW concentrations [48,49,50,51,52,53].

The overall efficacy of the treatments was ranked based on their ability to inhibit oxidation, with BHT at 200 ppm and 100 ppm leading the group, followed by EOSW 0.50%, which outperformed EOSW 0.10%. Specifically, BHT at 200 and 100 ppm exhibited the highest overall performance in both primary and secondary oxidation inhibition. EOSW 0.50% ranked second, effectively delaying oxidation processes similar to or surpassing low-dose BHT. EOSW 0.10% also showed strong performance, though slightly less effective than EOSW 0.50%. BHT 50 ppm ranked next, followed by EOSW 0.05%. Potassium sorbate (across all doses) showed the weakest overall efficacy, closely mirroring the control group, and providing minimal protection against oxidation. The control group exhibited the least efficacy, with the highest rates of both primary and secondary oxidation metabolites. These results highlight that EOSW at concentrations of 0.10% and 0.50% serves as a promising natural alternative to synthetic antioxidants like BHT, particularly in delaying both primary and secondary oxidation processes.

## 4. Conclusions

This study demonstrates a sustainable valorization pathway for onion solid waste (OSW) through optimized pressurized liquid extraction, spray-drying encapsulation, and application in a lipid-rich food system. Encapsulated OSW (EOSW) retained high levels of polyphenols, exhibited strong antioxidant activity, and performed effectively as a natural preservative in mayonnaise, showing comparable or even superior efficacy to synthetic BHT. Beyond its functional benefits, EOSW represents an environmentally responsible approach by converting an agri-food by-product into a value-added ingredient, aligning with the growing demand for natural, clean-label alternatives. These findings highlight EOSW as a promising candidate to replace synthetic preservatives, while future research should explore its application across diverse food systems and assess consumer acceptance.

## Figures and Tables

**Figure 1 foods-14-03583-f001:**
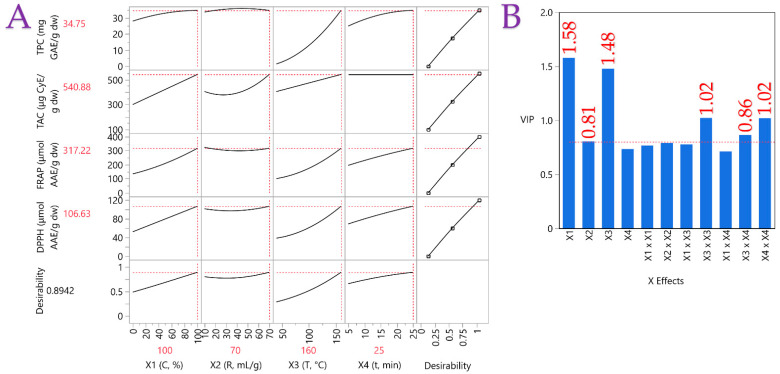
Plot (**A**) shows the optimization of the PLE process for OSW extracts using a partial least squares (PLS) prediction profiler combined with a desirability function under extrapolation control. Plot (**B**) displays the Variable Importance Plot (VIP), which ranks the predictor variable in the PLE model; the red dashed line indicates the 0.8 significance level.

**Figure 2 foods-14-03583-f002:**
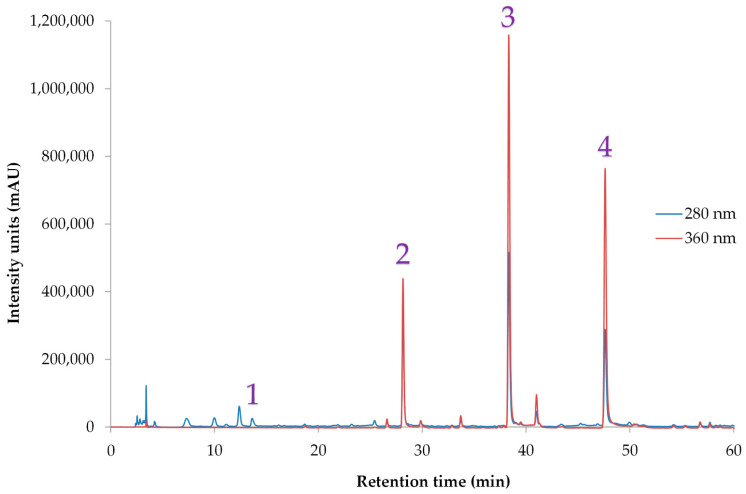
Representative HPLC chromatograms of the encapsulated OSW extract recorded at 280 nm and 360 nm, highlighting the detected polyphenolic compounds: 1—protocatechuic acid, 2—unknown flavonoid, 3—spiraeoside (quercetin 4′-*O*-glucoside), 4—quercetin.

**Table 1 foods-14-03583-t001:** Levels of independent variables included in the experimental design.

Independent Variables	Code Units	Coded Variable Level		
		−1	0	1
Aqueous ethanol concentration (*C*, % *v*/*v*)	*X* _1_	0	50	100
Liquid-to-solid ratio (*R*, mL/g) [equivalent to solvent-to-feed ratio, S/F]	*X* _2_	10	40	70
Extraction temperature (*T*, °C)	*X* _3_	40	100	160
Extraction time (*t*, min)	*X* _4_	5	15	25

**Table 2 foods-14-03583-t002:** Ingredients of mayonnaise.

Ingredient	Amount (g)	Percentage (%)	Purpose
Sunflower oil	220	71.2	Primary fat base
Olive oil	30	9.7	Extra flavor & fats
Egg yolk	17	5.5	Emulsifier
Mustard	15	4.9	Stabilizer & taste
Vinegar (6%)	13	4.2	Acidity
Lemon juice	12	3.9	Fresh acidity & brightness
Salt	1.5	0.5	Enhances flavor
Pepper powder	0.5	0.2	Adds subtle spice

Fat content per 100 g of product: From this mix, the total fat comes to around 255 g/300 g product ≈ 85 g fat per 100 g of mayonnaise.

**Table 3 foods-14-03583-t003:** Mayonnaise samples tested for their oxidative stability during a period of 14 days.

Sample No	Mayonnaise (g)	Encapsulated Extract (g)	BHT (ppm)	Potassium Sorbate (ppm)
1	100	-	-	-
2	100	0.05	-	-
3	100	0.1	-	-
4	100	0.5	-	-
5	100	-	50	-
6	100	-	100	-
7	100	-	200	-
8	100	-	-	250
9	100	-	-	500
10	100	-	-	1000

**Table 4 foods-14-03583-t004:** Experimental findings for the four investigated independent variables and the dependent variables’ responses to the PLE technique.

Design Point	Independent Variables				Actual PLE Responses *			
	*C* (%) (*X*_1_)	*R* (mL/g) (*X*_2_)	*T* (°C) (*X*_3_)	*t* (min) (*X*_4_)	TPC	TAC	FRAP	DPPH
1	100 (1)	70 (1)	160 (1)	25 (1)	34.20	511.16	286.84	94.01
2	50 (0)	40 (0)	100 (0)	15 (0)	13.71	220.34	39.25	19.78
3	100 (1)	40 (0)	68.8 (−0.52)	25 (1)	5.72	306.04	100.69	37.23
4	0 (−1)	40 (0)	40 (−1)	5 (−1)	2.19	122.44	17.72	9.15
5	0 (−1)	10 (−1)	160 (1)	12.5 (−0.25)	7.96	125.87	63.80	22.13
6	50 (0)	10 (−1)	100 (0)	5 (−1)	7.27	277.10	64.24	23.11
7	0 (−1)	40 (0)	160 (1)	25 (1)	30.85	165.66	130.12	46.83
8	50 (0)	40 (0)	100 (0)	15 (0)	12.13	216.99	38.77	19.34
9	0 (−1)	70 (1)	100 (0)	15 (0)	7.58	439.14	79.13	35.42
10	50 (0)	70 (1)	40 (−1)	25 (1)	8.07	291.22	59.91	29.42
11	50 (0)	40 (0)	40 (−1)	15 (0)	12.52	265.56	88.86	35.56
12	50 (0)	70 (1)	160 (1)	5 (−1)	10.91	359.28	81.04	40.68
13	50 (0)	40 (0)	100 (0)	15 (0)	10.79	204.38	38.68	19.27
14	100 (1)	40 (0)	160 (1)	5 (−1)	27.96	422.35	199.29	66.51
15	100 (1)	10 (−1)	40 (−1)	15 (0)	12.13	246.41	110.76	30.87
16	100 (1)	70 (1)	40 (−1)	5 (−1)	14.06	430.02	93.15	36.19
17	100 (1)	10 (−1)	160 (1)	25 (1)	33.82	443.05	352.14	112.62
18	0 (−1)	10 (−1)	40 (−1)	25 (1)	3.42	174.05	21.16	9.26

* Values represent the mean of triplicate determinations; TPC, total polyphenol content (in mg GAE/g dw); TAC, total anthocyanin content (μg CyE/g dw); FRAP, ferric reducing antioxidant power (in μmol AAE/g dw); DPPH, antiradical activity (in μmol AAE/g dw).

**Table 5 foods-14-03583-t005:** Analysis of variance (ANOVA) for the quadratic polynomial response surface model applied to the PLE technique.

Factor	TPC	TAC	FRAP	DPPH
Stepwise regression coefficients				
Intercept	12.53 *	240.2 *	50.66 *	24.19 *
*X*_1_—ethanol concentration	5.418 *	82.67 *	54.48 *	15.75 *
*X*_2_—liquid-to-solid ratio	0.566	68.17 *	−1.17	3.476
*X*_3_—temperature	7.278 *	30.6	48.29 *	16.06 *
*X*_4_—extraction time	3.821 *	-	25.85 *	8.392 *
*X* _1_ ^2^	−1.26	-	16.87	-
*X* _1_ *X_2_*	−2.39 *	-	−25.7	−8.56 *
*X* _2_ ^2^	−1.62	81.28 *	21.48	6.834
*X* _1_ *X* _3_	3.231 *	37.98	31.09 *	9.77 *
*X* _2_ *X* _3_	-	-	−11	−4.07
*X* _3_ ^2^	6.257 *	-	41.46 *	14.01 *
*X* _1_ *X* _4_	−6.3 *	-	-	-
*X* _2_ *X* _4_	2.11 *	-	-	-
*X* _3_ *X* _4_	6.324 *	-	28.68 *	8.331 *
*X* _4_ ^2^	−1.71	-	-	-
*ANOVA*				
*F-*value (model)	84.41	12.06	17.02	17.43
*F-*value (lack of fit)	0.518	56.22	11,446	1303
*p-*Value (model)	0.0003 *	0.0002 *	0.0012 *	0.0005 *
*p-*Value (lack of fit)	0.6586 ^ns^	0.0176 *	<0.0001 *	0.0008 *
*R* ^2^	0.996	0.834	0.969	0.961
Adjusted *R*^2^	0.985	0.765	0.912	0.906
RMSE	1.274	57.7	26.77	8.437
MR	14.2	290.1	103.6	38.2
CV	72.28	41.04	87.08	72.15
DF (total)	17	17	17	17

* The values significantly affected responses at a probability level of 95% (*p* < 0.05). TPC, total polyphenol content; TAC, total anthocyanin content; FRAP, ferric reducing antioxidant power; DPPH, antiradical activity; ns marks “non-significant results”; *F*-value, test for comparing model variance with residual (error) variance; *p*-Value, probability of seeing the observed *F*-value if the null hypothesis is true; RMSE, root mean square error; MR, mean of response; CV, coefficient of variation; DF, degrees of freedom.

**Table 6 foods-14-03583-t006:** Maximum predicted responses and optimum extraction conditions for the dependent variables.

Parameters	Independent Variables				Desirability	Stepwise Regression
	** *C* ** ** (%) (*X*_1_)**	** *R* ** ** _L/S_ ** ** (mL/g) (*X*_2_)**	***T*** **(°C) (*X*_3_)**	***t*** **(min) (*X*_4_)**		
TPC (mg GAE/g dw)	76	53	160	25	0.9912	35.69 ± 2.74
TAC (μg CyE/g dw)	100	70	160	-	0.9623	540.88 ± 81.24
FRAP (μmol AAE/g dw)	100	10	160	25	0.8765	356.66 ± 61.13
DPPH (μmol AAE/g dw)	100	10	160	25	0.9213	112.5 ± 18.53

**Table 7 foods-14-03583-t007:** The partial least squares (PLS) prediction profiler determined the maximum desirability for all variables under optimal extraction conditions for the PLE onion solid waste extract.

Parameters	Independent Variables				PLS Regression	PLE Experimental Values
	*C* (%) (*X*_1_)	*R* (mL/g) (*X*_2_)	*T* (°C) (*X*_3_)	*t* (min) (*X*_4_)		
TPC (mg GAE/g dw)	100	70	160	25	34.75	37.02 ± 1.18
TAC (μg CyE/g dw)					540.88	592.73 ± 31.41
FRAP (μmol AAE/g dw)					317.22	291.15 ± 17.76
DPPH (μmol AAE/g dw)					106.63	98 ± 6.57

**Table 8 foods-14-03583-t008:** Individual polyphenolic compounds of onion solid wastes, followed by encapsulation in gum arabic. Calibration curve equations for each compound as analyzed via HPLC-DAD are also included.

A/A	Polyphenolic Compound	C (mg/g)	C (mg/L)	LOD (mg/L)	LOQ (mg/L)	Equation	R^2^
1	Protocatechuic acid	0.34 ± 0.01	34.43 ± 0.9	7.47	22.64	y = 21,282.98x − 63,918.97	0.977
2	Unknown flavonoid ^1^	4.41 ± 0.31	441.21 ± 30.89	2.03	6.16	y = 45,367.87x − 56,445.13	0.999
3	Spiraeoside	9.73 ± 0.42	973.14 ± 41.85	9.41	28.51	y = 16,762.58x + 77,649.53	0.972
4	Quercetin	9.07 ± 0.5	906.92 ± 49.88	2.03	6.16	y = 45,367.87x − 56,445.13	0.999
	Total identified	23.56 ± 1.24	2355.7 ± 123.51				
	Spiraeoside/Quercetin ratio	1.07					

^1^ Quantified as quercetin equivalents.

**Table 9 foods-14-03583-t009:** Changes in pH of mayonnaise samples over 14-day storage.

Treatment	pH			
	Day 1	Day 3	Day 7	Day 14
Control	3.56 ± 0.12 ^A,B-a^	3.59 ± 0.07 ^A-a^	3.65 ± 0.22 ^A-a^	3.76 ± 0.09 ^A,B-a^
EOSW 0.05% (*w*/*w*)	3.6 ± 0.1 ^A,B-a^	3.63 ± 0.12 ^A-a^	3.69 ± 0.24 ^A-a^	3.8 ± 0.19 ^A,B-a^
EOSW 0.10% (*w*/*w*)	3.48 ± 0.19 ^A,B-a^	3.51 ± 0.26 ^A-a^	3.57 ± 0.19 ^A-a^	3.69 ± 0.12 ^A,B-a^
EOSW 0.50% (*w*/*w*)	3.36 ± 0.13 ^B-a^	3.39 ± 0.22 ^A-a^	3.45 ± 0.13 ^A-a^	3.56 ± 0.19 ^B-a^
BHT 50 ppm	3.46 ± 0.23 ^A,B-a^	3.49 ± 0.12 ^A-a^	3.55 ± 0.17 ^A-a^	3.66 ± 0.11 ^A,B-a^
BHT 100 ppm	3.54 ± 0.18 ^A,B-a^	3.57 ± 0.24 ^A-a^	3.63 ± 0.2 ^A-a^	3.74 ± 0.18 ^A,B-a^
BHT 200 ppm	3.52 ± 0.1 ^A,B-a^	3.55 ± 0.15 ^A-a^	3.61 ± 0.22 ^A-a^	3.72 ± 0.09 ^A,B-a^
Potassium sorbate 250 ppm	3.62 ± 0.2 ^A,B-a^	3.65 ± 0.12 ^A-a^	3.71 ± 0.13 ^A-a^	3.82 ± 0.2 ^A,B-a^
Potassium sorbate 500 ppm	3.64 ± 0.2 ^A,B-a^	3.67 ± 0.23 ^A-a^	3.73 ± 0.18 ^A-a^	3.84 ± 0.14 ^A,B-a^
Potassium sorbate 1000 ppm	3.87 ± 0.18 ^A-a^	3.9 ± 0.23 ^A-a^	3.96 ± 0.15 ^A-a^	4.07 ± 0.25 ^A-a^

Data are presented as mean ± standard deviation. Within each column, values bearing different superscript uppercase letters (A, B) differ significantly (*p* < 0.05) among treatments on the same storage day. Within each row, values bearing different superscript lowercase letters (a) differ significantly (*p* < 0.05) among storage days for the same treatment.

**Table 10 foods-14-03583-t010:** Color parameters (*L**, and *C**) of mayonnaise samples over 14-day storage.

Treatment	Day 1	Day 3	Day 7	Day 14
	** *L** **			
Control	90.7 ± 0.8 ^A,B-a^	88 ± 0.5 ^A,B-b^	82.6 ± 0.7 ^A,B-c^	73.2 ± 0.7 ^A,B-d^
EOSW 0.05% (*w*/*w*)	89.7 ± 0.5 ^A,B,C,D-a^	87.1 ± 0.7 ^A,B,C-b^	82.6 ± 0.5 ^A,B-c^	73.1 ± 0.5 ^A,B-d^
EOSW 0.10% (*w*/*w*)	88.6 ± 0.8 ^C,D,E-a^	86.2 ± 0.4 ^C,D-b^	81.3 ± 0.4 ^B,C,D-c^	72.8 ± 0.6 ^B-d^
EOSW 0.50% (*w*/*w*)	87.3 ± 0.5 ^E-a^	85.2 ± 0.9 ^D-b^	81.1 ± 0.8 ^B,C,D-c^	73.9 ± 0.5 ^A,B-d^
BHT 50 ppm	88.5 ± 0.6 ^D,E-a^	86.4 ± 0.6 ^B,C,D-b^	82.1 ± 0.4 ^A,B,C-c^	74.6 ± 0.6 ^A-d^
BHT 100 ppm	89.5 ± 0.9 ^A,B,C,D-a^	87 ± 0.4 ^A,B,C-b^	82 ± 0.7 ^A,B,C-c^	73.3 ± 0.6 ^A,B-d^
BHT 200 ppm	89.4 ± 0.4 ^B,C,D-a^	86.8 ± 0.9 ^B,C,D-b^	81.7 ± 0.4 ^A,B,C-c^	72.7 ± 0.6 ^B-d^
Potassium sorbate 250 ppm	90.5 ± 0.5 ^A,B,C-a^	87 ± 0.4 ^A,B,C-b^	79.9 ± 0.5 ^D-c^	67.5 ± 0.5 ^C-d^
Potassium sorbate 500 ppm	91.1 ± 0.7 ^A,B-a^	87.7 ± 0.6 ^A,B,C-b^	80.8 ± 0.7 ^C,D-c^	68.8 ± 0.3 ^C-d^
Potassium sorbate 1000 ppm	91.4 ± 0.5 ^A-a^	88.6 ± 0.4 ^A-b^	83 ± 0.4 ^A-c^	73.2 ± 0.7 ^A,B-d^
	** *C** **			
Control	12.9 ± 0.1 ^G-d^	17.2 ± 0.2 ^F-c^	25.8 ± 0.1 ^B,C-b^	40.9 ± 0.3 ^A-a^
EOSW 0.05% (*w*/*w*)	15.3 ± 0.1 ^E,F-d^	18.8 ± 0.1 ^E-c^	21.3 ± 0.2 ^E-b^	38.2 ± 0.3 ^C-a^
EOSW 0.10% (*w*/*w*)	17.6 ± 0.1 ^C-d^	20.3 ± 0.1 ^B,C-c^	25.8 ± 0.2 ^B,C-b^	35.4 ± 0.2 ^E-a^
EOSW 0.50% (*w*/*w*)	19.3 ± 0.1 ^B-d^	21.8 ± 0.1 ^A-c^	27.1 ± 0.2 ^A-b^	36.3 ± 0.4 ^D-a^
BHT 50 ppm	17.1 ± 0.1 ^D-d^	19.2 ± 0.1 ^D,E-c^	23.4 ± 0.2 ^D-b^	30.8 ± 0.2 ^F-a^
BHT 100 ppm	20.7 ± 0.2 ^A-d^	21.5 ± 0.1 ^A-c^	22.9 ± 0.2 ^D-b^	25.7 ± 0.2 ^H-a^
BHT 200 ppm	17.7 ± 0.1 ^C-d^	20.5 ± 0.2 ^B-c^	26.3 ± 0.2 ^B-b^	36.5 ± 0.4 ^D-a^
Potassium sorbate 250 ppm	17.2 ± 0.1 ^D-d^	20 ± 0.1 ^C-c^	25.5 ± 0.2 ^C-b^	35.3 ± 0.4 ^E-a^
Potassium sorbate 500 ppm	15.6 ± 0.1 ^E-d^	19.3 ± 0.1 ^D-c^	26.8 ± 0.2 ^A-b^	39.9 ± 0.4 ^B-a^
Potassium sorbate 1000 ppm	15 ± 0.2 ^F-d^	17.3 ± 0.2 ^F-c^	21.8 ± 0.2 ^E-b^	29.7 ± 0.2 ^G-a^

Data are expressed as mean ± standard deviation. Capital letters (i.e., A–H) indicate statistically significant differences (*p* < 0.05) between different treatments. Lowercase letters (i.e., a–d) indicate statistically significant differences (*p* < 0.05) among different storage days within the same treatment.

**Table 11 foods-14-03583-t011:** TPC of mayonnaise samples over 14-day storage.

Treatment	TPC (mg GAE/kg Mayonnaise)			
	Day 1	Day 3	Day 7	Day 14
Control	323.53 ± 21.35 ^A-a^	234.86 ± 4.7 ^A-b^	185.08 ± 7.77 ^C,D-c^	223.77 ± 8.06 ^A-b^
EOSW 0.05% (*w*/*w*)	168.96 ± 12.5 ^F-b,c^	160.81 ± 10.29 ^E-c^	229.43 ± 11.47 ^B-a^	197.51 ± 12.64 ^A,B-b^
EOSW 0.10% (*w*/*w*)	275.99 ± 19.6 ^B,C-a^	222.16 ± 9.33 ^A,B-b^	222.82 ± 6.91 ^B-b^	156.45 ± 4.54 ^C-c^
EOSW 0.50% (*w*/*w*)	224.75 ± 13.93 ^D,E-a^	192.04 ± 5.38 ^C,D-b^	124.23 ± 3.23 ^E-c^	174.32 ± 10.46 ^B,C-b^
BHT 50 ppm	289.85 ± 13.62 ^A,B-a^	173.38 ± 9.71 ^D,E-c^	211.2 ± 9.93 ^B,C-b^	172.55 ± 3.8 ^B,C-c^
BHT 100 ppm	237.38 ± 13.29 ^C,D-b^	224.53 ± 4.72 ^A,B-b^	304.26 ± 22.82 ^A-a^	198.98 ± 14.33 ^A,B-b^
BHT 200 ppm	185.24 ± 6.48 ^E,F-a^	190.63 ± 7.24 ^C,D-a^	170.15 ± 6.64 ^D-a^	179.31 ± 10.94 ^B,C-a^
Potassium sorbate 250 ppm	230.08 ± 16.57 ^D-a^	210.93 ± 9.49 ^A,B,C-a^	229.46 ± 4.59 ^B-a^	171.5 ± 3.94 ^B,C-b^
Potassium sorbate 500 ppm	225.48 ± 9.47 ^D,E-a^	210.49 ± 11.58 ^A,B,C-a,b^	184.27 ± 13.08 ^C,D-b^	182.22 ± 12.76 ^B,C-b^
Potassium sorbate 1000 ppm	228.02 ± 7.98 ^D-a^	198.93 ± 14.12 ^B,C,D-b^	179.63 ± 13.47 ^C,D-b^	178.76 ± 6.97 ^B,C-b^

Data are expressed as mean ± standard deviation. Capital letters (i.e., A–F) indicate statistically significant differences (*p* < 0.05) between different treatments. Lowercase letters (i.e., a–c) indicate statistically significant differences (*p* < 0.05) among different storage days within the same treatment.

**Table 12 foods-14-03583-t012:** DPPH radical scavenging activity of mayonnaise samples over 14-day storage.

Treatment	DPPH (μmol AAE/kg Mayonnaise)			
	Day 1	Day 3	Day 7	Day 14
Control	244.84 ± 12.73 ^B-b^	231.58 ± 6.48 ^B-b^	299.55 ± 20.37 ^D,E-a^	166.7 ± 9.5 ^D,E-c^
EOSW 0.05% (*w*/*w*)	215.11 ± 13.77 ^B-b^	159.41 ± 8.93 ^D-c^	372.11 ± 19.72 ^A,B-a^	208.6 ± 9.8 ^B-b^
EOSW 0.10% (*w*/*w*)	340 ± 8.84 ^A-b^	290.75 ± 15.99 ^A-c^	401.13 ± 21.26 ^A-a^	305.85 ± 10.09 ^A-b,c^
EOSW 0.50% (*w*/*w*)	238.66 ± 15.04 ^B-b^	203.28 ± 11.99 ^B,C-c^	291.9 ± 16.05 ^D,E-a^	222.64 ± 6.01 ^B-b,c^
BHT 50 ppm	180.85 ± 6.15 ^C-c^	216.78 ± 6.72 ^B,C-b^	298.68 ± 17.92 ^D,E-a^	205.6 ± 6.37 ^B-b,c^
BHT 100 ppm	223.92 ± 11.87 ^B-b^	161.42 ± 3.55 ^D-c^	274.89 ± 11 ^E-a^	197.17 ± 13.8 ^B,C-b^
BHT 200 ppm	245.04 ± 11.03 ^B-b^	208.06 ± 12.48 ^B,C-c^	322.64 ± 10 ^C,D-a^	176.6 ± 5.65 ^C,D-d^
Potassium sorbate 250 ppm	223.44 ± 10.05 ^B-b^	236.49 ± 13.24 ^B-b^	296.28 ± 10.07 ^D,E-a^	141.72 ± 5.1 ^E-c^
Potassium sorbate 500 ppm	169.42 ± 4.74 ^C-c^	219.42 ± 14.92 ^B-b^	293.28 ± 6.45 ^D,E-a^	155.83 ± 10.13 ^D,E-c^
Potassium sorbate 1000 ppm	221.35 ± 13.72 ^B-b^	184.35 ± 13.83 ^C,D-c^	345.22 ± 7.94 ^B,C-a^	173.63 ± 10.94 ^C,D-c^

Data are expressed as mean ± standard deviation. Capital letters (i.e., A–E) indicate statistically significant differences (*p* < 0.05) between different treatments. Lowercase letters (i.e., a–d) indicate statistically significant differences (*p* < 0.05) among different storage days within the same treatment.

**Table 13 foods-14-03583-t013:** DPPH^•^ radical scavenging activity of mayonnaise samples over 14-day storage.

Treatment	DPPH^•^ (μmol TEAC/kg Mayonnaise)			
	Day 1	Day 3	Day 7	Day 14
Control	1202.61 ± 80.57 ^D,E-a^	1082.35 ± 80.09 ^C,D,E-a^	901.95 ± 39.69 ^D,E-b^	721.57 ± 20.93 ^F,G-c^
EOSW 0.05% (*w*/*w*)	1227.4 ± 88.37 ^C,D,E-a^	1166.03 ± 62.97 ^B,C,D-a,b^	1043.29 ± 49.03 ^C,D-b^	859.18 ± 31.79 ^E,F-c^
EOSW 0.10% (*w*/*w*)	1209.01 ± 45.94 ^D,E-a^	1184.82 ± 87.68 ^B,C,D-a^	1147.7 ± 40.17 ^C-a^	1088.1 ± 30.47 ^C-a^
EOSW 0.50% (*w*/*w*)	1302 ± 39.06 ^B,C,D-a^	1263.96 ± 49.29 ^B,C-a^	1171.79 ± 52.73 ^C-a,b^	1041.6 ± 76.04 ^C,D-b^
BHT 50 ppm	1426.65 ± 28.53 ^B,C-a^	1311.6 ± 93.12 ^B-a^	1141.32 ± 59.35 ^C-b^	927.32 ± 40.8 ^D,E-c^
BHT 100 ppm	2000.08 ± 134.01 ^A-a^	1900.07 ± 133.01 ^A-a^	1760.07 ± 66.88 ^B-a,b^	1500.05 ± 93 ^B-b^
BHT 200 ppm	2127.49 ± 53.19 ^A-a^	2063.66 ± 45.4 ^A-a,b^	1978.57 ± 49.46 ^A-b^	1808.37 ± 56.06 ^A-c^
Potassium sorbate 250 ppm	1146.64 ± 83.7 ^D,E-a^	1008.19 ± 57.47 ^D,E-a^	802.65 ± 43.34 ^E-b^	573.33 ± 19.49 ^H-c^
Potassium sorbate 500 ppm	1475.12 ± 79.66 ^B-a^	1327.62 ± 73.02 ^B-a^	1106.34 ± 28.76 ^C-b^	811.32 ± 21.91 ^E,F-c^
Potassium sorbate 1000 ppm	1019.23 ± 54.02 ^E-a^	937.69 ± 18.75 ^E-a^	815.38 ± 50.55 ^E-b^	662.5 ± 48.36 ^G,H-c^

Data are expressed as mean ± standard deviation. Capital letters (i.e., A–H) indicate statistically significant differences (*p* < 0.05) between different treatments. Lowercase letters (i.e., a–c) indicate statistically significant differences (*p* < 0.05) among different storage days within the same treatment.

**Table 14 foods-14-03583-t014:** Peroxide value of mayonnaise samples over 14-day storage.

Treatment	PV (μmol H_2_O_2_/kg Mayonnaise)			
	Day 1	Day 3	Day 7	Day 14
Control	61.97 ± 4.03 ^A,B,C-c^	68.16 ± 4.57 ^A,B-c^	92.95 ± 2.88 ^A-b^	111.54 ± 5.69 ^A-a^
EOSW 0.05% (*w*/*w*)	56.78 ± 3.92 ^C,D-c^	61.32 ± 3.92 ^B,C-c^	76.65 ± 2.53 ^C-b^	90.85 ± 4 ^C-a^
EOSW 0.10% (*w*/*w*)	51.43 ± 1.39 ^D-c^	52.97 ± 2.91 ^C-c^	59.14 ± 1.54 ^D-b^	66.85 ± 1.34 ^D-a^
EOSW 0.50% (*w*/*w*)	61.63 ± 4.62 ^A,B,C-c^	64.71 ± 2.59 ^A,B-c^	77.03 ± 2.39 ^C-b^	89.36 ± 3.75 ^C-a^
BHT 50 ppm	66.64 ± 3.6 ^A,B-c^	71.31 ± 1.78 ^A-c^	86.63 ± 4.68 ^A,B-b^	99.96 ± 6.4 ^A,B,C-a^
BHT 100 ppm	3.83 ± 0.29 ^E-c^	4.02 ± 0.16 ^D-b,c^	4.59 ± 0.32 ^E-a,b^	5.17 ± 0.32 ^E-a^
BHT 200 ppm	4.42 ± 0.13 ^E-b^	4.56 ± 0.34 ^D-b^	5.08 ± 0.26 ^E-a,b^	5.53 ± 0.35 ^E-a^
Potassium sorbate 250 ppm	57.55 ± 2.88 ^B,C,D-c^	64.45 ± 4.77 ^A,B-c^	80.56 ± 5.48 ^B,C-b^	97.83 ± 2.35 ^B,C-a^
Potassium sorbate 500 ppm	66.98 ± 4.49 ^A-c^	73.68 ± 5.23 ^A-c^	90.42 ± 3.62 ^A-b^	107.17 ± 7.5 ^A,B-a^
Potassium sorbate 1000 ppm	62.22 ± 3.05 ^A,B,C-c^	67.82 ± 2.51 ^A,B-c^	80.89 ± 3.64 ^B,C-b^	96.44 ± 5.11 ^B,C-a^

Data are expressed as mean ± standard deviation. Capital letters (i.e., A–E) indicate statistically significant differences (*p* < 0.05) between different treatments. Lowercase letters (i.e., a–c) indicate statistically significant differences (*p* < 0.05) among different storage days within the same treatment.

**Table 15 foods-14-03583-t015:** TBARS of mayonnaise samples over 14-day storage.

Treatment	TBARS (μmol MDAE/kg Mayonnaise)			
	Day 1	Day 3	Day 7	Day 14
Control	0.36 ± 0.02 ^C-d^	0.54 ± 0.02 ^B,C-c^	0.65 ± 0.03 ^B,C-b^	0.9 ± 0.02 ^B-a^
EOSW 0.05% (*w*/*w*)	0.39 ± 0.01 ^B,C-d^	0.54 ± 0.02 ^B-c^	0.62 ± 0.02 ^B,C,D-b^	0.77 ± 0.02 ^B,C,D-a^
EOSW 0.10% (*w*/*w*)	0.38 ± 0.01 ^B,C-c^	0.46 ± 0.02 ^C,D-b^	0.5 ± 0.01 ^F-b^	0.58 ± 0.04 ^F,G-a^
EOSW 0.50% (*w*/*w*)	0.41 ± 0.03 ^B-c^	0.54 ± 0.02 ^B,C-b^	0.6 ± 0.02 ^C,D,E-b^	0.74 ± 0.03 ^C,D,E-a^
BHT 50 ppm	0.35 ± 0.02 ^C-c^	0.48 ± 0.03 ^B,C,D-b^	0.55 ± 0.03 ^D,E,F-b^	0.67 ± 0.03 ^D,E,F-a^
BHT 100 ppm	0.39 ± 0.03 ^B,C-c^	0.47 ± 0.03 ^B,C,D-b^	0.53 ± 0.02 ^E,F-b^	0.63 ± 0.03 ^E,F,G-a^
BHT 200 ppm	0.38 ± 0.01 ^B,C-c^	0.44 ± 0.03 ^D-b,c^	0.48 ± 0.04 ^F-a,b^	0.53 ± 0.04 ^G-a^
Potassium sorbate 250 ppm	0.35 ± 0.01 ^C-d^	0.53 ± 0.04 ^B,C-c^	0.67 ± 0.04 ^B,C-b^	0.88 ± 0.05 ^B-a^
Potassium sorbate 500 ppm	0.38 ± 0.01 ^B,C-d^	0.55 ± 0.01 ^B-c^	0.68 ± 0.03 ^B-b^	0.87 ± 0.06 ^B,C-a^
Potassium sorbate 1000 ppm	0.47 ± 0.01 ^A-d^	0.66 ± 0.04 ^A-c^	0.8 ± 0.02 ^A-b^	1.04 ± 0.07 ^A-a^

Data are expressed as mean ± standard deviation. Capital letters (i.e., A–G) indicate statistically significant differences (*p* < 0.05) between different treatments. Lowercase letters (i.e., a–d) indicate statistically significant differences (*p* < 0.05) among different storage days within the same treatment.

## Data Availability

The original contributions presented in this study are included in the article. Further inquiries can be directed to the corresponding author.

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
