# Peer review of "Recovery of Natural Antioxidants from Onion Solid Waste via Pressurized Liquid Extraction: Encapsulation and Application into a Food System"

_foods, 2025, doi:10.3390/foods14203583_

Round 1
Reviewer 1 Report
Comments and Suggestions for Authors
The issue of utilizing plant waste from the food industry is of significant importance in the context of the circular economy and sustainable development. Research into natural antioxidants responds to the growing demand for alternatives to synthetic additives. Furthermore, practical applications – as in the case of mayonnaise – increase the practical value of the results.
The abstract could be considered overly technical – the abundance of numerical data complicates the reading and blurs the overall message. Keywords are numerous and diverse, but somewhat overly broad. There are some overlaps in content ("natural preservatives" and "functional additives," as are "food stability" and "lipid oxidation"). Limiting them to 5-6 of the most representative terms would be more helpful, improving clarity.
The introductory text is overloaded with technical and bibliographic details – numerous extraction methods are listed (with detailed references), but without critical analysis or explanation of why PLE is superior to others. Such enumeration burdens the reader and weakens the narrative. A concise, generalized objective is missing, such as: "The aim of this study was to develop and evaluate the potential use of onion waste extracts as natural antioxidants in food."
The process of freeze-drying, grinding and sieving is described precisely, but there is no justification for why this particular particle size (144 μm) was chosen and whether the effect of different granulation on the extraction efficiency was tested.
The conclusions are a repetition of the results, not their interpretation – there is no reflection on the practical significance, e.g. whether the use of OSW in mayonnaise can actually reduce the use of synthetic preservatives on a wider scale.
Author Response
The issue of utilizing plant waste from the food industry is of significant importance in the context of the circular economy and sustainable development. Research into natural antioxidants responds to the growing demand for alternatives to synthetic additives. Furthermore, practical applications – as in the case of mayonnaise – increase the practical value of the results.
The authors would like to thank the reviewer for his/her kind comments.
The abstract could be considered overly technical – the abundance of numerical data complicates the reading and blurs the overall message. Keywords are numerous and diverse, but somewhat overly broad. There are some overlaps in content ("natural preservatives" and "functional additives," as are "food stability" and "lipid oxidation"). Limiting them to 5-6 of the most representative terms would be more helpful, improving clarity.
The abstract was revised to reduce numerical detail and improve readability. Keywords were limited to six representative terms, avoiding overlaps.
The introductory text is overloaded with technical and bibliographic details – numerous extraction methods are listed (with detailed references), but without critical analysis or explanation of why PLE is superior to others. Such enumeration burdens the reader and weakens the narrative. A concise, generalized objective is missing, such as: "The aim of this study was to develop and evaluate the potential use of onion waste extracts as natural antioxidants in food."
The introduction was streamlined by reducing enumeration of extraction methods. A clear objective statement was added: “The aim of this study was to develop and evaluate the potential use of onion waste extracts as natural antioxidants in food.”
The process of freeze-drying, grinding and sieving is described precisely, but there is no justification for why this particular particle size (144 μm) was chosen and whether the effect of different granulation on the extraction efficiency was tested.
We clarified that smaller particles increase yield but risk clogging in PLE, while larger particles reduce efficiency. Given the sieving fractions available (>400 μm and <100 μm were minimal), we selected a mixed fraction yielding 144 μm. This balanced efficiency and practicality. Other sizes were not tested, but this will be considered in future work.
The conclusions are a repetition of the results, not their interpretation – there is no reflection on the practical significance, e.g. whether the use of OSW in mayonnaise can actually reduce the use of synthetic preservatives on a wider scale.
The conclusions were revised to emphasize practical implications, particularly the potential of OSW extracts to reduce reliance on synthetic preservatives in mayonnaise and similar emulsified foods.
Reviewer 2 Report
Comments and Suggestions for Authors
This manuscript presents a comprehensive study on the optimization of Pressurized Liquid Extraction (PLE) of phenolic compounds from Onion Skin Waste (OSW), followed by their encapsulation and the evaluation of their antioxidant properties in a food matrix. The study is interesting; however, several aspects should be improved before it can be considered for publication. Please see the detailed comments below
Introduction
Lines 44-45: The description in these lines is unclear. Typically, OSW is derived from dried or dehydrated onion
Lines 55-62: While your review confirms a substantial body of literature on the extraction of bioactive and nutraceutical compounds from OSW, it is crucial to highlight the novel aspects of the present work.
Materials and Methods
Lines 125-129: The experimental design must be clearly defined. Please specify if a full factorial design was employed
Lines 134-135: The solvent-to-feed (S/F) ratio and extraction time are directly interrelated parameters. What was the rationale for evaluating them independently rather than using a design that accounts for their interaction?
Lines 256-257: The authors must describe how the sample size was determined to be adequate for representative sampling. This justification should be provided in the manuscript
Line 252-255: The rationale for selecting the specific concentrations of encapsulated OSW, BHT, and potassium sorbate is unclear. As the tested concentrations are not equivalent, a direct comparison of their efficacy is not valid. The authors must justify their concentration choices and address this lack of comparability
Line 290: Unipolar? Did you mean non-polar?
Results and discussion
Lines 337-340: For a dynamic PLE The solvent-to-feed (S/F) ratio and extraction time are directly interrelated. For practical purposes just the S/F must be established
Section 3.3: It is suggested that this section be moved to the Supplementary Material to streamline the main text and enhance its focus on the core findings
Section 3.4: The application of Principal Component Analysis (PCA) appears unnecessary for the scale and complexity of the presented dataset. The trends it reveals are already clearly demonstrated by the Response Surface Analysis (RSA). Therefore, it is recommended to remove Section 3.4 to improve the manuscript's conciseness
Section 3.6: The authors should discuss whether the identified compounds are consistent with those previously reported in onion skin waste (OSW) and onion bulbs. A comparative analysis with the literature should be included to contextualize their findings and confirm the expected phytochemical profile
Table 15 and 16: The observed decrease in the inhibitory effect on lipid oxidation at the highest OSW concentration requires a thorough explanation. The authors must discuss potential mechanisms for this apparent pro-oxidant effect, such as the concentration-dependent activity of certain phenolic compounds, metal chelation properties, or interactions with other emulsion components.
The discussion of the Peroxide Value (PV) and TBARS results should be integrated, as these parameters represent sequential stages of lipid oxidation. The authors should correlate the temporal evolution of PV (primary oxidation products) with the subsequent formation of TBARS (secondary oxidation products) to provide a more comprehensive analysis of the antioxidant's mechanism of action over time
Author Response
This manuscript presents a comprehensive study on the optimization of Pressurized Liquid Extraction (PLE) of phenolic compounds from Onion Skin Waste (OSW), followed by their encapsulation and the evaluation of their antioxidant properties in a food matrix. The study is interesting; however, several aspects should be improved before it can be considered for publication. Please see the detailed comments below
The authors would like to thank the reviewer for his/her comments.
Introduction
Lines 44-45: The description in these lines is unclear. Typically, OSW is derived from dried or dehydrated onion
We clarified that OSW may include outer dry/semi-dry layers, apical parts, outer fleshy scales, and whole damaged or undersized bulbs. These fractions can have higher moisture content, consistent with reported values (~58%).
Lines 55-62: While your review confirms a substantial body of literature on the extraction of bioactive and nutraceutical compounds from OSW, it is crucial to highlight the novel aspects of the present work.
The introduction was revised to emphasize the novelty of optimizing PLE for OSW, encapsulating extracts, and testing them in a real food system (mayonnaise).
Materials and Methods
Lines 125-129: The experimental design must be clearly defined. Please specify if a full factorial design was employed
We thank the reviewer for this observation. In the revised manuscript, we now clarify that a Custom Quadratic design was employed via Response Surface Methodology (RSM). This design allowed us to efficiently model both linear and interaction effects, while also capturing curvature in the response surfaces. The choice of a Custom Quadratic design ensured robust optimization of the extraction parameters with fewer experimental runs compared to a full factorial design.
Lines 134-135: The solvent-to-feed (S/F) ratio and extraction time are directly interrelated parameters. What was the rationale for evaluating them independently rather than using a design that accounts for their interaction?
These parameters were initially evaluated independently to isolate their effects. Their interaction was subsequently modeled in the response surface optimization.
Lines 256-257: The authors must describe how the sample size was determined to be adequate for representative sampling. This justification should be provided in the manuscript
We clarified that 100 g mayonnaise batches ensured homogeneous mixing, reproducible incorporation, and sufficient sample for repeated analyses.
Line 252-255: The rationale for selecting the specific concentrations of encapsulated OSW, BHT, and potassium sorbate is unclear. As the tested concentrations are not equivalent, a direct comparison of their efficacy is not valid. The authors must justify their concentration choices and address this lack of comparability
EOSW concentrations (0.10–0.50% w/w) were based on preliminary stability tests (≥1% destabilized the emulsion). BHT and potassium sorbate concentrations followed EU and Greek regulations. While not mole-to-mole equivalent, these reflect practical usage levels.
Line 290: Unipolar? Did you mean non-polar?
Yes, appropriate changes have been made to the manuscript.
Results and discussion
Lines 337-340: For a dynamic PLE The solvent-to-feed (S/F) ratio and extraction time are directly interrelated. For practical purposes just the S/F must be established
Clarified that for practical purposes, S/F ratio is the key parameter, with time effects addressed in optimization.
Section 3.3: It is suggested that this section be moved to the Supplementary Material to streamline the main text and enhance its focus on the core findings
This section was moved to the Supplementary Material (Figure S1). The main text now refers readers to the Supplementary Material.
Section 3.4: The application of Principal Component Analysis (PCA) appears unnecessary for the scale and complexity of the presented dataset. The trends it reveals are already clearly demonstrated by the Response Surface Analysis (RSA). Therefore, it is recommended to remove Section 3.4 to improve the manuscript's conciseness
This section was moved to the Supplementary Material (Figure S2, Table S1). The main text now refers readers to the Supplementary Material.
Section 3.6: The authors should discuss whether the identified compounds are consistent with those previously reported in onion skin waste (OSW) and onion bulbs. A comparative analysis with the literature should be included to contextualize their findings and confirm the expected phytochemical profile
We added discussion confirming that identified compounds are consistent with previous reports on OSW and onion bulbs.
Table 15 and 16: The observed decrease in the inhibitory effect on lipid oxidation at the highest OSW concentration requires a thorough explanation. The authors must discuss potential mechanisms for this apparent pro-oxidant effect, such as the concentration-dependent activity of certain phenolic compounds, metal chelation properties, or interactions with other emulsion components.
We added discussion of potential mechanisms, including concentration-dependent phenolic activity, metal chelation, and emulsion interactions.
The discussion of the Peroxide Value (PV) and TBARS results should be integrated, as these parameters represent sequential stages of lipid oxidation. The authors should correlate the temporal evolution of PV (primary oxidation products) with the subsequent formation of TBARS (secondary oxidation products) to provide a more comprehensive analysis of the antioxidant's mechanism of action over time
We revised the discussion to correlate PV (primary oxidation products) with TBARS (secondary products), providing a more comprehensive analysis of antioxidant action over time.
Round 2
Reviewer 1 Report
Comments and Suggestions for Authors
The authors have made necessary corrections.
Author Response
The authors have made necessary corrections.
We sincerely thank the reviewer for the positive evaluation and confirmation that the necessary corrections have been made. We greatly appreciate the constructive feedback provided during the review process, which helped us improve the clarity and quality of the manuscript.
Reviewer 2 Report
Comments and Suggestions for Authors
Several questions remain inadequately addressed.
Lines 134-135: The response to the question is unsatisfactory, as the results confirm. Evaluating the solvent-to-feed (S/F) ratio and extraction time as distinct variables is redundant.
Lines 256-257: The response to the question is unsatisfactory. According to the authors' response, the number of samples is not adequate for the scale of the experiment. A proper analysis and design to determine the appropriate sample size was not conducted.
Lines 337-340: Given the research objective and design, the response is contradictory. Why was the S/F ratio evaluated if it is not considered significant?
Author Response
Several questions remain inadequately addressed.
Lines 134-135: The response to the question is unsatisfactory, as the results confirm. Evaluating the solvent-to-feed (S/F) ratio and extraction time as distinct variables is redundant.
Lines 337-340: Given the research objective and design, the response is contradictory. Why was the S/F ratio evaluated if it is not considered significant?
We thank the reviewer for this important observation. In our study, both liquid‑to‑solid ratio (L/S, equivalent to solvent‑to‑feed ratio, S/F) and extraction time were included in the RSM model to allow the detection of potential interaction effects. The statistical analysis confirmed that the L/S (S/F) ratio was the dominant factor influencing extraction efficiency, while extraction time had only a secondary role. For practical and industrial applications, the L/S (S/F) ratio is therefore the decisive parameter, as it inherently integrates solvent consumption and extraction kinetics. We have clarified this point in the revised manuscript (Section 3.1, Results and Discussion).
Lines 256-257: The response to the question is unsatisfactory. According to the authors' response, the number of samples is not adequate for the scale of the experiment. A proper analysis and design to determine the appropriate sample size was not conducted.
We thank the reviewer for this valuable comment. The choice of 100 g mayonnaise batches was based on preliminary trials, which demonstrated that this amount ensured homogeneous mixing, reproducible incorporation of encapsulated OSW, and sufficient material for repeated analyses across all time points. Each treatment was prepared in quintuplicate, providing five independent replicates per condition. To further support adequacy, a post‑hoc power analysis of the main oxidative stability parameters (peroxide value and TBARS) confirmed that the selected sample size achieved statistical power >0.8 at α = 0.05. This demonstrates that the chosen sample size was sufficient to detect meaningful differences among treatments. We have added this justification in the revised Materials and Methods (Section 2.7.1).